# Using VAEs to Learn Latent Variables: Observations on Applications in cryo-EM

## Abstract

Variational autoencoders (VAEs) are a popular generative model used to approximate distributions. The encoder part of the VAE is used in amortized learning of latent variables, producing a latent representation for data samples. Recently, VAEs have been used to characterize physical and biological systems. In this case study, we qualitatively examine the amortization properties of a VAE used in biological applications. We find that in this application the encoder bears a qualitative resemblance to a more traditional explicit representation of latent variables.

## 1 Introduction

Variational Autoencoders (VAEs) provide a deep learning method for efficient approximate inference for problems with continuous latent variables. A brief reminder about VAEs is presented in Section 2.1; a more complete description can be found, inter alia, in Kingma & Welling (2014; 2019); Amos (2022); Cremer et al. (2018); Kim & Pavlovic (2021); Doersch (2021). Since their introduction, VAEs have found success in a wide variety of fields. Recently, they have been used in scientific applications and physical systems (Zhong et al., 2021a; Sandfort et al., 2021; Takeishi & Kalousis, 2021; Baur et al., 2020; Donnat et al., 2022).

Given a set of data $x = \{x_i\}$, VAEs simultaneously learn an encoder $\text{Enc}_\xi$ that expresses a conditional distribution $q_\xi(z|x)$ of a latent variable $z_i$ given a sample $x_i$, and a decoder $\text{Dec}_\theta$ which expresses the conditional distribution $p_\theta(x|z)$. They are trained using empirical samples to approximate the distribution $p_\theta(x, z)$.

In this work we focus on the properties of the encoder distribution $q_\xi(z|x)$ that arise as an approximation of the distribution $p_\theta(z|x)$. A single encoder $q_\xi(z|x)$ is optimized to be able to produce the distribution of latent variable $z$ for any input $x$, which is a form of amortization. Intuitively, one might expect that the encoder $q_\xi(z|x)$ would generalize well to plausible inputs that it has not encountered during the optimization/training procedure. Indeed, this generalization is observed in many applications, and the ability of the encoder to compute the latent variables for new unseen data points is used in some applications. In addition, the variational construction sidesteps a statistical problem by marginalizing over the latent variables to approximate the maximum-likelihood estimator (MLE) for some parameters $\theta$ of the distribution $p_\theta(x, z)$, rather than $\theta$ *and* the latent variables $z_i$ associated with each sample $x_i$. In the latter case, the number of variables grows with the number of samples and the estimates of $p_\theta(x, z)$ may not converge to the true solution.

We present a qualitative case study of the amortization in VAEs in a physical problem, looking at a VAE applied to the problem of continuous heterogeneity in cryo-electron microscopy (cryo-EM), implemented in CryoDRGN (Zhong et al., 2021a). We examine the hypothesis that the encoder in this VAE generalizes well to previously unseen data, and we compare the use of a VAE to the use of an explicit variational estimation of the distribution of the latent variables. In order to study the generalization in a realistic environment, we exploit well-known invariances and approximate invariances in cryo-EM data to produce natural tests.

Our case study suggests that in this case the encoder does not seem to generalize well; this can arguably be interpreted as a form of overfitting of the data. Furthermore, we find that using explicit latent variables

(variational approximations and arguably also explicit values) yields qualitatively similar (and arguably better) estimates than using the encoder in this test case.

We would like to clarify that the purpose of this case study is not to criticize the work in Zhong et al. (2021a), but rather to draw attention to possibly surprising properties of VAEs in some applications and to the parallels between VAEs and explicit latent variables in these applications. The phenomena observed here are not exhibited in every application; for completeness we present in Appendix B similar experiments on classic VAEs trained on the MNIST dataset (LeCun et al., 2010).

The work is loosely inspired by the work in Zhang et al. (2017); one could conceivably draw some conceptual parallels between the over-fitting demonstrated there and some of the experiments presented in this paper in the context of VAEs.

The code used for this paper has been made available at

`https://github.com/danieledelberg/ExplicitLatentVariables`.

## 2 Preliminaries

### 2.1 Variational Inference and Variational Autoencoders

In this section we provide a brief reminder of VAEs, adapted from the formulation in Kingma & Welling (2019).

Figure 1 illustrates the standard VAE neural network, combining an encoder $\text{Enc}_\xi$ and a decoder $\text{Dec}_\theta$. For the observations $\{x_1, \ldots, x_n\}$, we aim to infer a distribution $p_\theta(x)$. We denote by $z_1, \ldots, z_n$ the latent variables; these are an unobserved component of the model. The parameters of the generative model for the decoder are denoted by $\theta$. The marginal distribution $p_\theta(x)$ over the observed variables is given by

$$p_\theta(x) = \int_z p_\theta(x, z) dz \tag{1}$$

The classic Maximum-Likelihood Estimation problem gives the following optimization problem to solve:

$$\theta^* = \arg\max_\theta L(x, \theta) = \arg\max_\theta \sum_{i=1}^n \log p(x_i|\theta) \tag{2}$$

Or equivalently,

$$\theta^* = \arg\max_\theta \sum_{i=1}^n \log \int_{z_i} p_\theta(x|z) p_\theta(z) dz \tag{3}$$

However, this integration in Equation 3 is intractable, and thus it is desirable to have an approximation to the distribution $p_\theta(z|x)$. We utilize an encoder model $q_\xi(z|x)$, where $\xi$ are the parameters of this inference model, called the variational parameters. We intend to optimize these parameters to best approximate

$$q_\xi(z|x) \approx p_\theta(z|x) \tag{4}$$

The distribution $q_\xi(z|x)$ can be parameterized by a neural network. A popular choice is a neural network encoder $\text{Enc}_\xi(x_i)$ which produces a mean $\mu_i$ and variance $\sigma_i$ of a multivariate Gaussian distribution for any input $x_i$,

$$\text{Enc}_\xi(x_i) = (\mu_i, \log \sigma_i) \tag{5}$$

$$q_\xi(z_i|x_i) = \mathcal{N}(z_i; \mu_i, \text{diag}(\sigma_i)) \tag{6}$$

Since the function $\text{Enc}_\xi$ shares its variational parameters $\xi$ across all $x_i$'s, this process is called amortized variational inference. This is in contrast to fitting explicit distributions $q_{\xi_1}(z_1|x_1), \ldots, q_{\xi_n}(z_n|x_n)$ for each datapoint separately with parameters $\xi_1, \ldots, \xi_n$ that are not shared across distributions (the latter is what we will later do in the experiment in Section 3.2). In many applications, the amortized learning of shared

encoder variables reduces the computational cost; the encoder often generalizes so that it can produce a reasonable approximate $q_\xi(z|\tilde{x})$ for samples $\tilde{x}$ that were not used in training.

In the decoder, the parameters $\theta$ are fitted to approximate the true distribution $p^*(x|z)$ in a similar manner as we described for the case of the encoder. In order to fit the distribution $q_\xi(z|x)$, one must formulate an alternative objective function, since the original maximum-likelihood formulation in Equation 2 does not include a distribution $q$. We write the objective function as:

$$L(x, \theta) = \mathbb{E}_{q_\xi(z|x)} \left[ \log p_\theta(x, z) - \log q_\xi(z|x) \right] + \mathbb{E}_{q_\xi(z|x)} \left[ \log q_\xi(z|x) - \log p_\theta(z|x) \right] \tag{7}$$

The second expectation term is the Kullback-Leibler (KL) divergence between $q_\xi(z|x)$ and $p_\theta(z|x)$, written as $D_{KL}(q_\xi(z|x) \| p_\theta(z|x))$, which is non-negative. The first expectation term is called the evidence-based lower bound (ELBO) $\mathcal{L}_{\theta,\xi}(x)$. The ELBO loss serves as a lower-bound to $L(x, \theta)$ and we attempt to find a series of distributions $q_1, \ldots, q_n$ that will maximize this lower bound. For completeness, a reminder of the derivation of the ELBO is presented in Appendix A. When we use the ELBO as an objective function for optimization, we often write it as

$$\mathcal{L}_{\theta,\xi}(x) = \mathbb{E}_{q_\xi(z|x)} \left[ \log p_\theta(x|z) \right] - D_{KL}(q_\xi(z|x) \| p_\theta(z)) \tag{8}$$

Typically, we choose $q_\xi(z|x)$ to be a multivariate Gaussian distribution (as in Equation 6) and $p_\theta(z) \sim \mathcal{N}(0, 1)$ so that the KL Divergence term can be written analytically. The first term on the right hand side is the expected reconstruction error.

The amortization gap (Cremer et al., 2018) characterizes the difference between the optimized $q_{\hat{\xi}}$ and the best possible posterior from the set of all possible distributions parameterized by the network $\text{Enc}_\xi$ with respect to the ELBO.

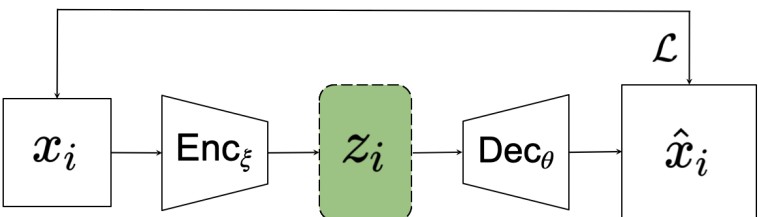

Figure 1: Experimental setup of a standard VAE. $x_i$ is a datapoint used as input, $z_i$ is the associated latent space point, $\hat{x}_i$ is the reconstructed output of $x_i$. $\mathcal{L}$ represents the calculation of the ELBO between $x$ and $\hat{x}_i$.

## 2.2 Case Study: Cryo-Electron Microscopy

Cryo-EM is an imaging technology that uses an electron beam to create a tomographic projection of a biological macromolecule frozen in a layer of vitreous ice. The location and orientation of particles within the ice is random and unknown. The aim of classic cryo-EM reconstruction is to estimate the 3D structure of the biomolecule using the 2D particle images. Due to the radiation sensitivity of the biomolecules and lack of contrast enhancement agents, particle images often have a very low signal-to-noise (SNR) ratio. Advances in recent years have facilitated reconstruction of these 3D structures at resolutions approaching 1-2Å (Renaud et al., 2018). One of the actively studied questions in cryo-EM is that of of heterogeneity: how to infer multiple 3D structures (molecular conformations) from measurements of mixtures of different macro-molecules. Despite the success in analyzing mixtures of distinct discrete conformations, the analysis of continuums of conformations (e.g. flexible molecules) has been an open problem until recent years, and it remains an important active area of research (Toader et al., 2022). A very brief formulation of the problem

is presented in the next subsection. A more detailed description of the cryo-EM problem formulation can be found, inter alia, in Bendory et al. (2019).

### 2.2.1 Mathematical Formulation and Image Model

Let $V : \mathbb{R}^3 \to \mathbb{R}$ be the 3D structure we want to estimate. In the simplified cryo-EM model, we assume that the particle image $x_i$ is created by rotating $V$ by some rotation $\omega_i \in SO(3)$, performing a tomographic projection, shifting by some translation $t_i \in \mathbb{R}^2$, convolving with the contrast transfer function of the microscope $h_i$, and adding noise. This can be written as

$$x_i = h_i * (T_{t_i} P R_{\omega_i} V) + \epsilon_i \tag{9}$$

where $R_{\omega_i}$ is the rotation operator that applies rotation $\omega_i$ to the volume, $T_{t_i}$ is the operator for translation by $t_i$, $P$ is the tomographic projection operator, and $\epsilon_i$ is the frequency-dependent noise. In the heterogeneous problem, the single volume $V$ is replaced with a function $V_z$, where $z$ is the latent conformation variable, which is often embedded in a lower-dimensional space in the continuous formulation introduced in Lederman & Singer (2017); Lederman et al. (2020). In the discrete formulation of the heterogeneity problem, $z = \{1, \dots, K\}$ represents the $K$ discrete volumes of the structure (Bendory et al., 2019).

### 2.2.2 Case Study VAE: CryoDRGN

One of the methods for analyzing continuous heterogeneity in cryo-EM is CryoDRGN, a customized VAE specifically designed for generating 3D biomolecule structures from pre-processed micrographs (Zhong et al., 2021a;b). The network utilizes an initial pose set, composed of rotations $\omega_1, \dots, \omega_n$, translations $t_1, \dots, t_n$, and contrast transfer functions $h_1, \dots, h_n$, provided by an upstream homogeneous reconstruction, typically via an expectation-maximization algorithm. CryoDRGN's encoder provides a variational estimate $q_\xi(z_i|x_i)$ (as in Equation 6) in the form of the mean $\mu_i$ and variance $\sigma_i$ of a conditional distribution (as in Equation 5).

CryoDRGN's decoder is a specialized neural network-parameterized decoder which renders a slice $\hat{x}_i$ of volume $V_i$ given the latent variable $z_i$ provided by the encoder (and the parameters $\omega_i$, $t_i$ and $h_i$ provided by upstream algorithms); the interesting details of CryoDRGN's decoder are outside the scope of this paper. CryoDRGN's encoder is a standard fully connected MLP. The size of the network and the dimensions of the latent space are configurable. CryoDRGN's architecture is illustrated in Figure 2.

Our experiments are based on the original CryoDRGN network, with modification only to the encoder component of the network.

## 3 Methods and Results

Our experiments are based on modified versions of the CryoDRGN code; the original code is available at https://github.com/zhonge/cryodrgn. In order to make the comparison as informative as possible, we tried to minimize the modifications to the original code, rather than optimize the alternative setups. For similar reasons, we chose a well-studied dataset with well-studied parameters used in the CryoDRGN tutorial (Kinman et al., 2022) as discussed in Section 3.1. The run times of the different experiments in this section were virtually the same (see more details in Appendix D), which suggests similar computational costs for the different approaches.

Our naive null assumption is that the encoder utilizes the structure of the data to produce an informative latent space; we introduce a number of modifications to qualitatively test this hypothesis. In Section 3.1 we establish a baseline for our experiments by following a given tutorial of the CryoDRGN software package. In Section 3.2 we compare qualitatively the amortized encoder to a simple implementation of analogous explicit variational approximations of latent variables associated with each particle image separately. In Section 3.3 we investigate qualitatively the ability of the autoencoder to work without real information content. In Section 3.4 we qualitatively test the ability of the network to generalize to small perturbations in the data.

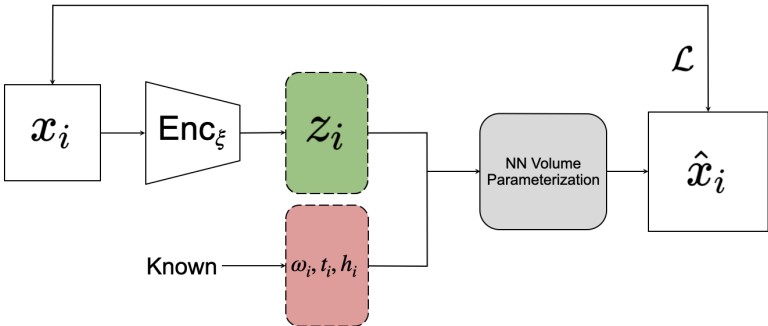

Figure 2: The CryoDRGN pipeline for heterogeneous reconstruction. $x_i$ is the cryo-EM image datapoint as input. $\omega_i$, $t_i$, $h_i$ are the given rotation, translation, and contrast transfer function of the input image. These variables with latent variable $z_i$ are used as input to the CryoDRGN volume decoder which then generates a projection image $\hat{x}_i$, which is compared to $x_i$ via the loss function $\mathcal{L}$ and parameters of the network are updated accordingly.

### 3.1 Baseline

As a baseline for our case-study we used the well-studied EMPIAR-10076 L17-Depleted 50S Ribosomal Intermediates dataset (Davis et al., 2016) with well-studied parameters for CryoDRGN, used in the CryoDRGN tutorial described in detail in Zhong & Bansal (2020). We chose to focus on the first part of the tutorial, where the particle images are downsampled to 128 x 128 and run through the initial training process. Following the tutorial recommendations, we ran the the network for 50 epochs with a learning rate of $10^{-4}$. The encoder and decoder were both composed of 3 hidden layers, each with 256 neurons. Per the defaults, we used a fully connected encoder architecture with ReLU activation and Fourier representation in the decoder. We did not use the built-in optional pose optimization methods.

As mentioned above, CryoDRGN uses estimates for the rotation, translation, and contrast transfer function for each particle that are provided by upstream algorithms, and focuses on the conformation variable $z$.

The results of the baseline experiment are illustrated in Figure 3. We note that different initialization of the algorithm leads to results that are different, but qualitatively comparable. The scatter plot represents the means of the latent variable $z_i$ for each particle image; the UMAP algorithm (McInnes et al., 2020) is used for two-dimensional visualization of the 8-dimensional latent variable. We use the latent variables associated with the first ten images as reference points (in red). In addition, we present some examples of 3-D conformation that the decoder produces from the latent variables of the first six particle images. We note that the UMAP algorithm is randomized algorithm, so results between runs differed slightly.

### 3.2 Experimental System: Variational Lookup Table

In this section we qualitatively compare the amortized encoder to a simple implementation of an analogous explicit variational approximations of latent variables associated with each particle image separately. We replaced the encoder deep network $\text{Enc}_\xi : x_i \rightarrow (\mu_i, \sigma_i)$ from Section 2.1 with a lookup table $\text{LT}_i : i \rightarrow (\mu_i, \sigma_i)$, to which we refer as a Variational Lookup Table (VLT). The VLT keeps track of an explicit mode $\mu_i$ and variance $\sigma_i$ for the latent variable for each individual image separately from the other images, whereas the encoder keeps a set of parameters $\xi$ that is shared across all the images.

In a classic VAE or in the CryoDRGN architecture, an index $i$ is used to choose a datapoint from the training set $x_i$ to feed-forward through the encoder, producing a parameterization of $q_\xi(z_i|x_i)$. The VLT chooses a row in a table associated with the index, which corresponds to a parameterization of the latent distribution $q$ for the $i$th particle image. The architecture is implemented by replacing the encoder model in the CryoDRGN

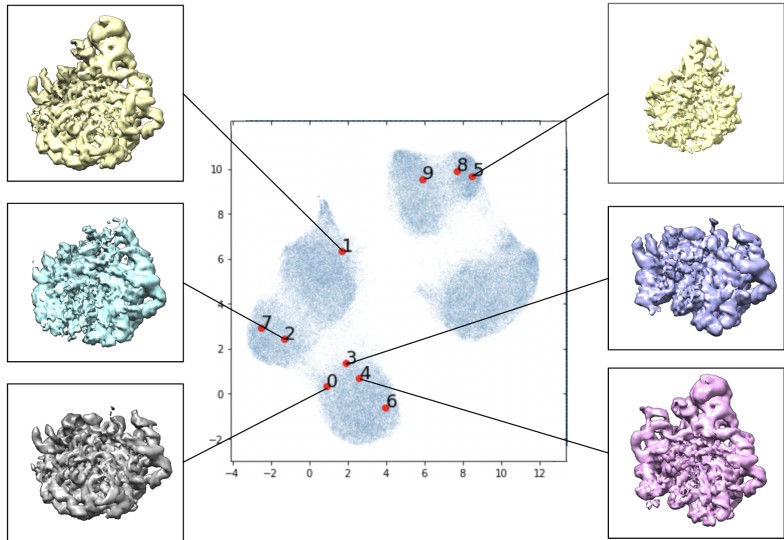

Figure 3: Low-resolution preliminary CryoDRGN results following the tutorial. Labeled locations in the learned latent space of the first ten images in the data set visualized with UMAP dimensionality reduction. Volumes associated with certain areas of the latent space are included.

software package with a Pytorch (Paszke et al., 2019) embedding table. In both architectures, during the optimization process, a value $z \sim q$ is sampled from $q$ for each particle image, and passed through the decoder. The entries in the table are optimized by standard backpropagation methods. The table can also be optimized using a separate optimization scheme or learning rate from the decoder.

This architecture shares some features with the tools described in Zadeh et al. (2021), such as the lack of an explicit encoder module, although our implementation allowed for some additional experimentation with optimization methods and has been adjusted to fit within the CryoDRGN pipeline. The architecture of the VLT is illustrated in Figure 4.

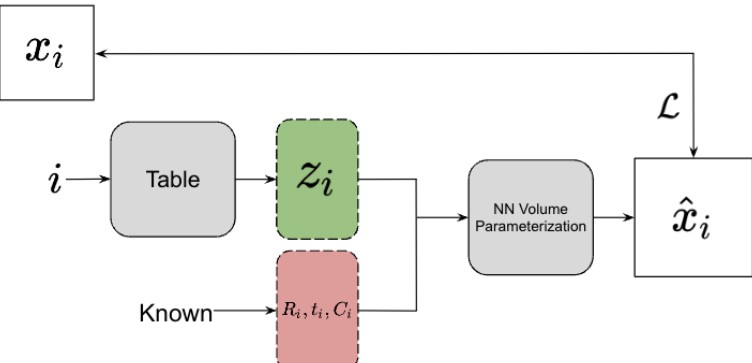

Figure 4: Experimental setup of the VLT. The architecture is similar to Figure 2, with the exception that the input $x_i$ to the network and its subsequent encoding with $\text{Enc}_\xi$ are replaced with the use of the index $i$ of the image as input to the latent space sampling function. The loss function $\mathcal{L}$ compares the image $x_i$ and the generated output of the network $\hat{x}_i$.

The result of the VLT experiment, using a normal Gaussian initialization, are presented in Figure 5. Additional results with different optimization techniques, as well as results where the VLT latent variables are fixed to those produced by the original CryoDRGN encoder are presented in Appendix C.

In the absence of established quantitative methods for comparison of heterogeneous structure analysis in cryo-EM (Toader et al., 2022), we resort to a qualitative examination. Qualitatively, the latent space structure and the volumes/conformations recovered by the VLT procedure agree with the original CryoDRGN results shown in Figure 3. The classification of individual particle images to specific conformations does not always agree with that produce in the baseline CryoDRGN run, but we note that this classification is not entirely consistent across CryoDRGN runs with different seeds and otherwise similar parameters.

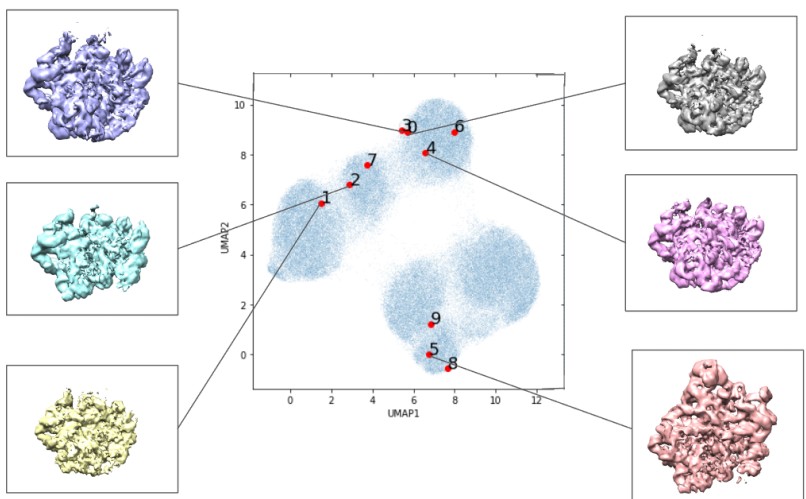

Figure 5: Random initialization of the latent space using the VLT architecture with latent space optimization. Labeled locations in the learned latent space of the first ten images in the data set visualized with UMAP dimensionality reduction. Volumes associated with certain areas of the latent space are included.

### 3.3 Experimental System: "Evil Twin"

To examine the relationship of the input to the latent variables via the encoder function, we employed a series of experiments we termed "Evil Twin," aiming to qualitatively examine the ability of the VAE to overfit particle images. In these experiments, we paired each observed particle image $x_i$ with some arbitrary "twin" $\tilde{x}_i$; the first experiment is a permutation test, where the twins are some different particle image from the dataset (no two particle images can have the same twin. The pairing is not symmetric; image $x_1$ may have $x_2$ as its evil twin, but $x_2$ may have $x_3$ as its evil twin, etc). Importantly, the assigned evil twin does not change during the optimization process. The $\tilde{x}_i$ was always used in the forward pass through the neural network in place of the non-evil image $x_i$, but the loss function was calculated against the image $x_i$. In other words, the encoder in the VAE is shown $\tilde{x}_i$, but the decoder is expected to produce $x_i$. A diagram of the architecture is shown in Figure 6. The result of the permutation evil twin experiment is presented in Figure 7, and results with a larger network are in Figure 8. The larger encoder network used 5 encoding layers with width 1024 each, compared to the original architecture of 3 encoding layers of width 256 each.

In the third evil-twin experiment we paired each particle image with a random noise image of the same dimension, where the mean and the variance of the noise were set to the empirical mean and variance of the dataset. We used the larger network size for this experiment, results are in Figure 9.

We find that the conformational variability revealed by the larger networks in the evil twin experiments in Figures 8 and 9 is qualitatively comparable to the baseline CryoDRGN in Figure 3. The standard-size network (identical to the network in the baseline) applied to our permutated evil twin setup in Figure 7 does

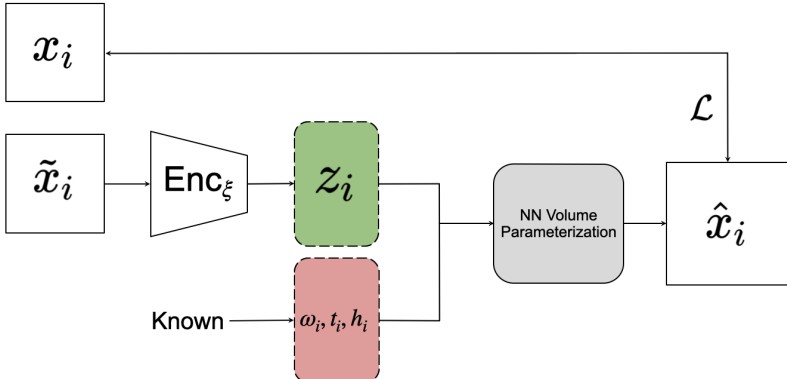

Figure 6: Experimental setup of the evil twin experiment. The architecture is similar to Figure 2, with the exception that the input to the network $x_i$ is instead the image's chosen evil-twin $\tilde{x}_i$. The loss function $\mathcal{L}$ compares the image $x_i$ and the generated output of the network $\hat{x}_i$.

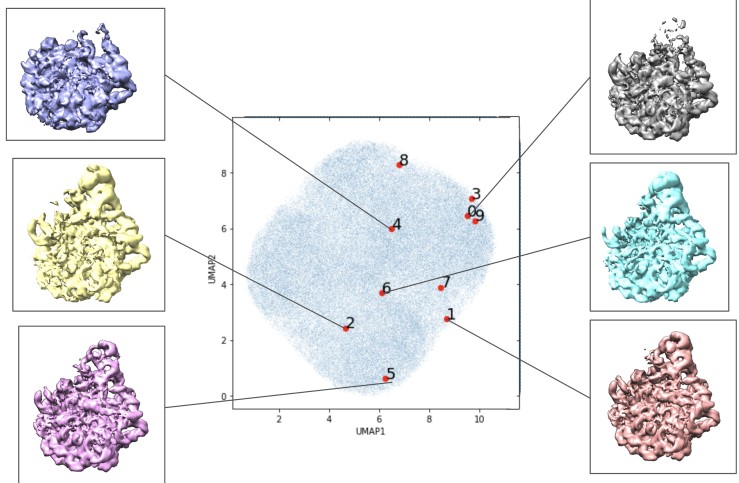

Figure 7: The evil twin experiment, permutation variant. Labeled locations in the learned latent space of the first set of images in the data set visualized in the 2-dimensional latent space. Volumes associated with certain areas of the latent space are included.

not yield distinct clusters as the baseline CryoDRGN setup in Figure 3; however, a closer examination of the actual volumes in Figure 7 suggests that it does reveal significant conformational variability.

## 3.4 Amortization Experiments

In many applications the encoder generalizes and can be used to approximate the conditional distribution $p(z|x)$ of samples $x$ not observed in the training phase. Evaluating this property in experimental cryo-EM data is challenging in the absence of a ground-truth. Fortunately, cryo-EM provides a natural experiment: since particles freeze in arbitrary orientations, the particles in the images can show up in random orientations and are not centered. This means that an in-plane rotated particle image should be assigned the same latent variable as the original image. Due to inaccuracy in cropping of particle images from the large micrograph produced by the microscope, small in-plane translations of the particle in the particle image should not lead to large changes in the latent variable.

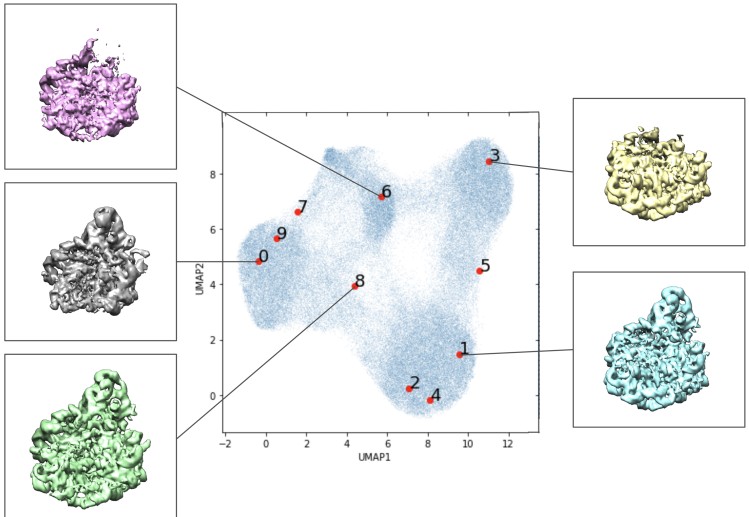

Figure 8: The evil twin experiment, permutation variant with a larger network. Labeled locations in the learned latent space of the first set of images in the data set visualized in the 2-dimensional latent space. Volumes associated with certain areas of the latent space are included.

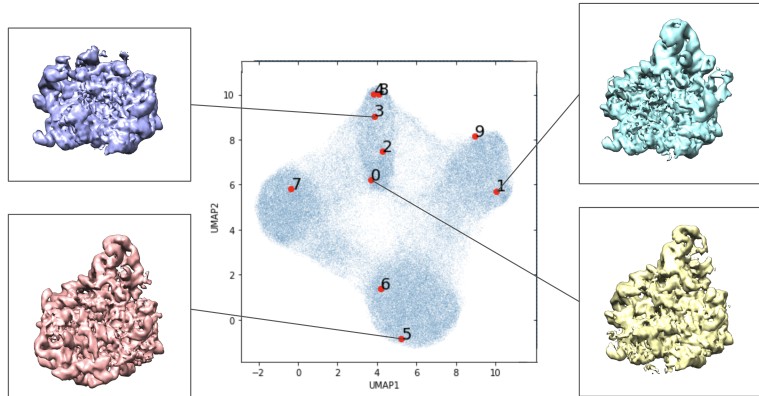

Figure 9: The evil twin experiment, noise variant with a larger network. Labeled locations in the learned latent space of the first set of images in the data set visualized in the 2-dimensional latent space. Volumes associated with certain areas of the latent space are included.

To test the hypothesized generalization and amortization properties of the VAE, we employed a series of experiments to augment the data. In these experiments we chose to train the network in a standard manner, and then shifted all of the training images by one pixel to the right, with the rightmost column pixels rolled over to the leftmost column of the image, to serve as test images. We also performed a series of analogous experiments with rotations by using training images rotated by 90° clockwise to serve as test images. If the encoder generalizes well to images that it has not seen before, we expect it to generalize well to invariances; for example, we would expect the rotated particle images to be assigned the same latent variables as the original particle image.

We introduced the shift on the naive assumption that the edges of the images consist of only noise, and that on 128x128 images a single pixel does not represent a large enough shift that should alter the center of the image nor the image content. Similarly, a rotation of 90° presents no issues of interpolation. Furthermore, because the translation and rotation of the image is fed after encoding the image to a latent space, we should expect that a translated or rotated image should result in a similar volume output from the decoder, and be mapped to a similar position in the latent space by the decoder, in keeping with the assumption of amortized learning of heterogeneous volumes.

We found that for both shifting the images by 1 pixel and rotating images by 90°, these augmentations appeared to have a significant impact on where the encoder mapped images to in the latent space. There appeared to be no qualitative agreement between $z_i$ values produced from encoding images and encoding their shifted or rotated variants. We found that shifted or rotated images still appeared to show heterogeneity and the various different structures produced were qualitatively similar across the whole latent space, but due to being mapped to very different points in the latent space, the shifted or rotated images were encoded and then decoded to create sometimes very qualitatively distinct volumes from those that the non-augmented versions produced. See Figure 10 for the shifted results, and Figure 11 for the rotated results. Some of the red labeled points appearing in the Figures also correspond to the labels in Figure 3, as the shifted and rotated results are from the same trained network as the tutorial.

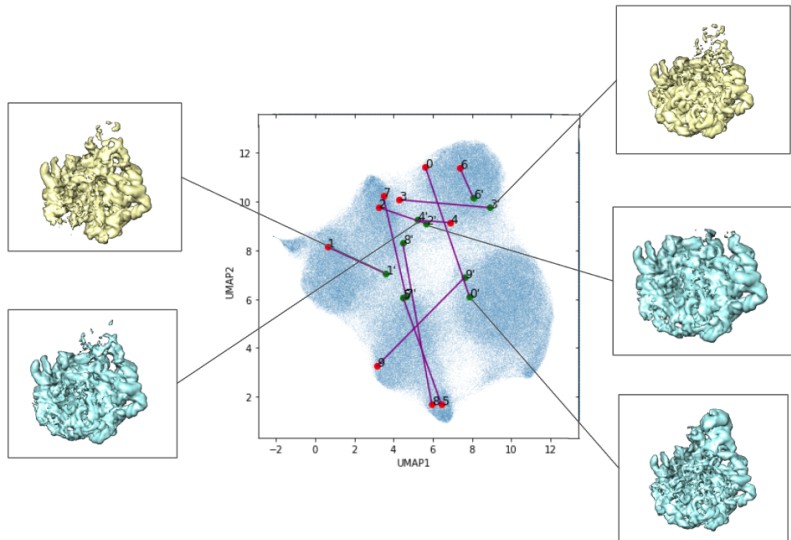

Figure 10: Shifted-by-1 images passed into a trained CryoDRGN network. Labeled locations in the learned latent space of the first set of images in the data set visualized in the 2-dimensional latent space. The first 10 training images are in red, the first 10 test images, shifted versions of the associated training images are in green and denoted with a '. Purple lines connect the labeled images and their associated shifted test image in the latent space. Volumes associated with certain areas of the latent space are included.

## 4   Discussion

The purpose of this case study was to examine if the common wisdom about VAEs holds in certain scientific applications. We based our case study on CryoDRGN, a modified VAE developed for cryo-EM applications. We limited ourselves to a very well-studied real-world experimental setup that has been used in tutorials for this software. Where we modified the setup or the data from the baseline setup, as described above, we tried to make the minimal necessary changes, rather than optimize the architecture and parameters.

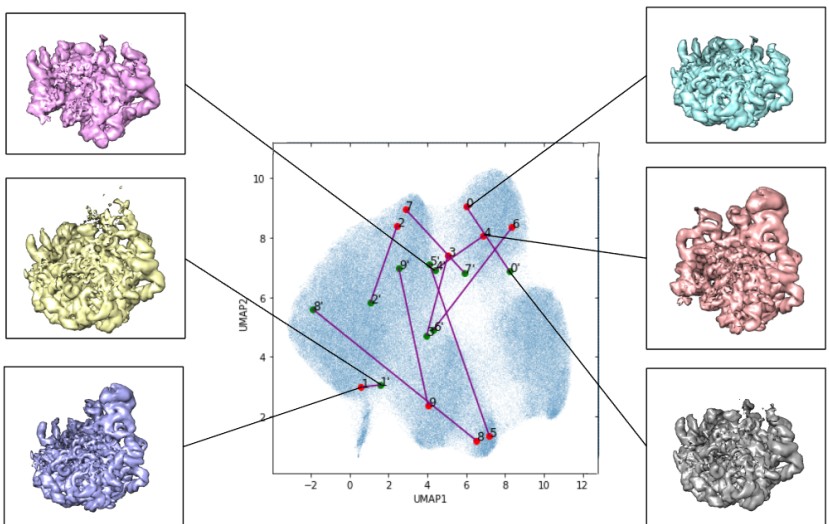

Figure 11: Rotated images passed into a trained CryoDRGN network. Labeled locations in the learned latent space of the first set of images in the data set visualized in the 2-dimensional latent space. The first 10 training images are in red, the first 10 test images, rotated versions of the associated training images are in green and denoted with a '. Purple lines connect the labeled images and their associated rotated test image in the latent space. Volumes associated with certain areas of the latent space are included.

In the experiment in Section 3.2, we used explicit variational approximations of latent variables (VLT) rather than an amortized encoder variational approximation. The results in the VLT setup are qualitatively similar to those in the original VAE setup. Clearly, the VLT does not have an immediate mechanism for generalization, suggesting that the amortization associated with the generalization in the VAE is not an important property in this implementation. The qualitative resemblance between the outputs of the VLT and the original VAE suggests that the encoder may not be the "secret sauce" that makes the VAE produce good results. The more traditional approaches to latent variables, evaluating them explicitly (in the broad sense, including sampling from their posterior for example), may be as applicable as the encoder approach.

In the experiments in Section 3.3, we replaced the input to the encoder with arbitrary images or even random noise; the original images are still used in the loss function in the comparison to the output of the encoder. Somewhat surprisingly, the algorithms still performs well (qualitatively), although this required a slightly larger network. In this case, there is no generalization that the encoder can do from one input to another (other than some level of restriction of the latent outputs that is applicable to all the inputs). One possible interpretation is that the encoder can effectively overfit the values of the latent variable to each individual image, effectively turning the encoder into something analogous to the explicit table in Section 3.2. We note that the SNR in cryo-EM data is often less than $1/10$, so noise dominates the images; we hypothesize (informally) that the encoder does not somehow "isolate" the signal in the images, but rather overfits to "noise features" as well.

The experiments in Section 3.4 examine the ability of the encoder to generalize. In the absence of ground-truth data, we use known invariances in the cryo-EM data to examine how well the encoder generalizes to augmented data, as a proxy for generalization to unseen test data. The encoder does not appear to generalize well to the augmented data, suggesting that it would also not generalize well to completely new unseen test data.

CryoDRGN does not enforce an invariance or approximate invariance with respect to rotations and translations in its network architecture. In principle, it is possible to build some invariance into the network architecture, enforcing encoders that are invariant to in-plane rotations for example. This invariance has not been implemented in CryoDRGN. It is not obvious if this would be beneficial to the primary goal of

recovering the different conformations, as we have not ruled out that the overfitting could be beneficial in some way in solving this problem. Furthermore, while such invariant network structures would solve the generalization problem demonstrated in our experiment (augmenting the images with rotated and translated images), it does not immediately imply that the network would generalize well to unseen particle images (truly new samples, as opposed to our augmented samples).

In Appendix B, we present analogous experiments with a classic MNIST VAE. The experiments are presented to provide some baseline in a more standard setup, and the results should be taken in this context. While there are some similarities to the case study, the results and applications are different in classic VAEs, represented by the MNIST example. The experiment in Section B.1.4 appears to indicate that explicit variables can be used in the MNIST VAE, but we did not examine in depth the generative properties of the model, and this experiment ignores the importance of the trained encoder itself as a tool that can be applied to unseen samples. MNIST images are centered and somewhat aligned, so augmentation based on translations or rotations of the MNIST images are not as good as those in cryo-EM for the purpose of characterizing the generalization. In short, our conclusions have limited applicability to classic VAE applications like MNIST.

The cryo-EM problem, and especially the conformational heterogeneity aspect of cryo-EM, has several special characteristics that might make it different from some other applications; some of these may apply to other scientific applications. One characteristic that stands out is the high level of noise in particle images. Intuitively, one can argue that without a very good implicit kernel, the high level of noise make every two images very different from each other even if their clean versions were identical; in other words, one could argue that the experiment in Section 3.3 where we pair each particle image with a random noise image, is not very far from the real setup. The encoder in the cryo-EM problem also has to "deal" with very high variability in the input: particle images that should be assigned the same conformation can be taken from different viewing direction, have different in-plane rotations and translation and may be subject to different filters. Furthermore, the determination of conformations may be a poorly conditioned: it may be difficult to tell the difference between different conformations from certain viewing directions even in the absence of noise.

It should also be noted that this case study does not apply to every possible implementation of VAEs in cryo-EM applications. Indeed, CryoAI (Levy et al., 2022a) appears to implement a VAE that determines the viewing direction of each particle image (but not the conformation) with good generalization properties. Given the limited intended scope of this case-study, and the fact that the code for CryoAI and the later CryoFIRE (Levy et al., 2022b) was not publicly available at the time of the writing, we did not expand our investigation.

## 5 Conclusion

VAEs are a powerful recent tool in scientific imaging applications. Our case study experiments demonstrate that the common wisdom about the properties of VAEs and encoders might not apply directly to every application, even when the VAE architecture appears to be very successful in practice. Furthermore, our results suggest that a VAE with an amortized encoder can behave surprisingly similarly to a more traditional explicit variable, which could point to directions of future work on where each approach is preferable.

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

## A    Derivation of the ELBO

For variational autoencoders, the objective to optimize is called the evidence-based lower bound (ELBO), sometimes called the variational lower bound. We choose an inference model $q_\xi(z|x)$ with variational pa-

rameters $\xi$. We follow the derivation in Kingma & Welling (2019).

$$\log p_\theta(x) = \mathbb{E}_{q_\xi(z|x)}[\log p_\theta(x)] \tag{10}$$

$$= \mathbb{E}_{q_\xi(z|x)}\left[\log\left[\frac{p_\theta(x,z)}{p_\theta(z|x)}\right]\right] \tag{11}$$

$$= \mathbb{E}_{q_\xi(z|x)}\left[\log\left[\frac{p_\theta(x,z)q_\xi(z|x)}{q_\xi(z|x)p_\theta(z|x)}\right]\right] \tag{12}$$

$$= \mathbb{E}_{q_\xi(z|x)}\left[\log\left[\frac{p_\theta(x,z)}{q_\xi(z|x)}\right]\right] + \mathbb{E}_{q_\xi(z|x)}\left[\log\left[\frac{q_\xi(z|x)}{p_\theta(z|x)}\right]\right] \tag{13}$$

Where we may write

$$D_{KL}(q_\xi(z|x)\|p_\theta(z|x)) = \mathbb{E}_{q_\xi(z|x)}\left[\log\left[\frac{q_\xi(z|x)}{p_\theta(z|x)}\right]\right] \tag{14}$$

$$\mathcal{L}_{\theta,\xi}(x) = \mathbb{E}_{q_\xi(z|x)}\left[\log\left[\frac{p_\theta(x,z)}{q_\xi(z|x)}\right]\right] \tag{15}$$

We may also write this as

$$\mathcal{L}_{\theta,\xi}(x) = \mathbb{E}_{q_\xi(z|x)}\left[\log[p_\theta(x|z)] + \log[p_\theta(z)] - \log[q_\xi(z|x)]\right] \tag{16}$$

$$= \mathbb{E}_{q_\xi(z|x)}\left[\log[p_\theta(x|z)]\right] - D_{KL}(q_\xi(z|x)\|p_\theta(z)) \tag{17}$$

## B  MNIST Experiments

### B.1  MNIST

We performed a similar set of experiments using a small network and the MNIST dataset (LeCun et al., 2010). These experiments are presented only as a better-known baseline, they do not exhibit all the same phenomena (see discussion, Section 4). We cleaned and preprocessed the dataset. We used Pytorch to construct an encoder with three hidden layers, width 256, ReLU activation, and used two separated layers at the end to create the $z_\mu$ and $z_\sigma$ variables. We then passed these two variables $z_\mu$ and $z_\sigma$ to a sampling function to get a resultant $z$ variable for each input in a batch:

$$\sigma = \exp(\frac{1}{2}z_\sigma) \tag{18}$$

$$\epsilon \sim \mathcal{N}(0,1) \tag{19}$$

$$z = z_\mu + \sigma \cdot \epsilon. \tag{20}$$

To create a random variable with mean $z_\mu$ and variance $\exp(z_\sigma)$. The decoder is two hidden layers, ReLU activation, with sigmoid activation for the final result.

The code is available on our Github repo.

#### B.1.1  Purpose

One advantage of the MNIST dataset over the CryoDRGN dataset besides being smaller and easier to work with is that we know what the resultant images "should" look like. In the case of the cryo-EM datasets, the images are often too noisy to know if we are looking at a "correct" encoding of the image, i.e. that an image from structure A is encoded into a $z$ value that produces structure A in the trained model. In MNIST, the images are only signal, no noise, so we know if the number in the image is correctly encoded and then decoded into the appropriate number.

### B.1.2 Standard VAE

The standard VAE produced results as we would expect: it got most of the images "correct" in that they are mapped by the decoder to an image that looks close enough to what the input looks like. We can see in the figures below that the model is run to approximate convergence, and the latent $z$ space is approximately clustered by number. For numbers like 0 and 1 we saw more separation than for numbers like 6 and 8, which was expected. We also generated a $z$ space that does not look very normally distributed, and has a few features that are very "non-normal" looking, like lines. Different runs could make the shapes look particularly sharp rather than rounded/normal shape that we expect from clusters in a normal distribution. We can also see that the network tended to show poor separation for distinguishing 4s vs 9s, 8s, 6s, and other numbers that were not as sharp as 1 or as rounded as 0, but rather had some "hybrid" structure in them. The results also appeared to be very sensitive to the random initializations, and there were a small number of cases where the results will not converge at all or will find very poor local minima they cannot escape. It was also very sensitive to learning rate, sometimes shooting off to very large values within a single batch. Loss function values can change by 20-30% between iterations regularly after convergence. See Figure 12 for these results, and Figure 13 for some example images and their reconstructions.

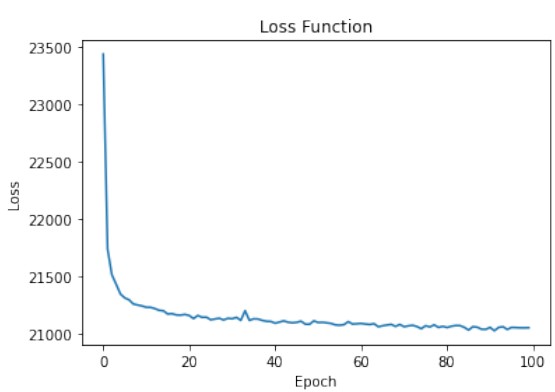
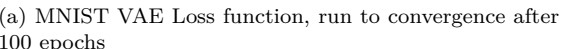
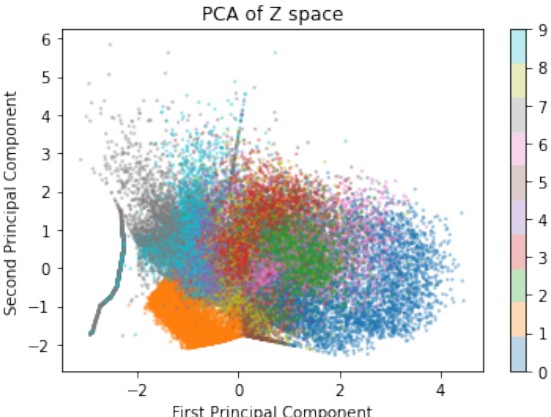

(a) MNIST VAE Loss function, run to convergence after 100 epochs

(b) The 2D latent space learning by the MNIST VAE, the Principal components are equivalent to the latent space since there is no dimensionality reduction. We can see the separation of some distinct numbers

Figure 12: The Standard MNIST VAE

### B.1.3 MNIST Amortization Experiments

For experiments involving testing amortization properties we chose to analyze shifting images. We trained the network on the standard dataset and then applied a transformation to the data to make it the new test dataset, which we ran through the network after the training procedure. We chose some number $s$ for shift size, either 1 or 5 in our experiments, and applied a roll operation to the right where the right-most column of the image was rolled over to the left-most column.

See Figures 14, 15 for results of the shifted-by-1 experiments, called Shift1. See Figures 16, 17 for results of the shifted-by-5 experiments, called Shift5.

### B.1.4 MNIST VLT

For the VLT experiments on MNIST data, we initialized the $z$ values in the latent space to zero, so all the $z_i$ started at the same point. Our latent $z$ space looked significantly more "normal" in distribution than the standard VAE, and we can see some more prominent clusters from 0 and 1 like in the VAE, although we also saw some more separation among the more "hybrid" numbers, such as 2, but in general many of the

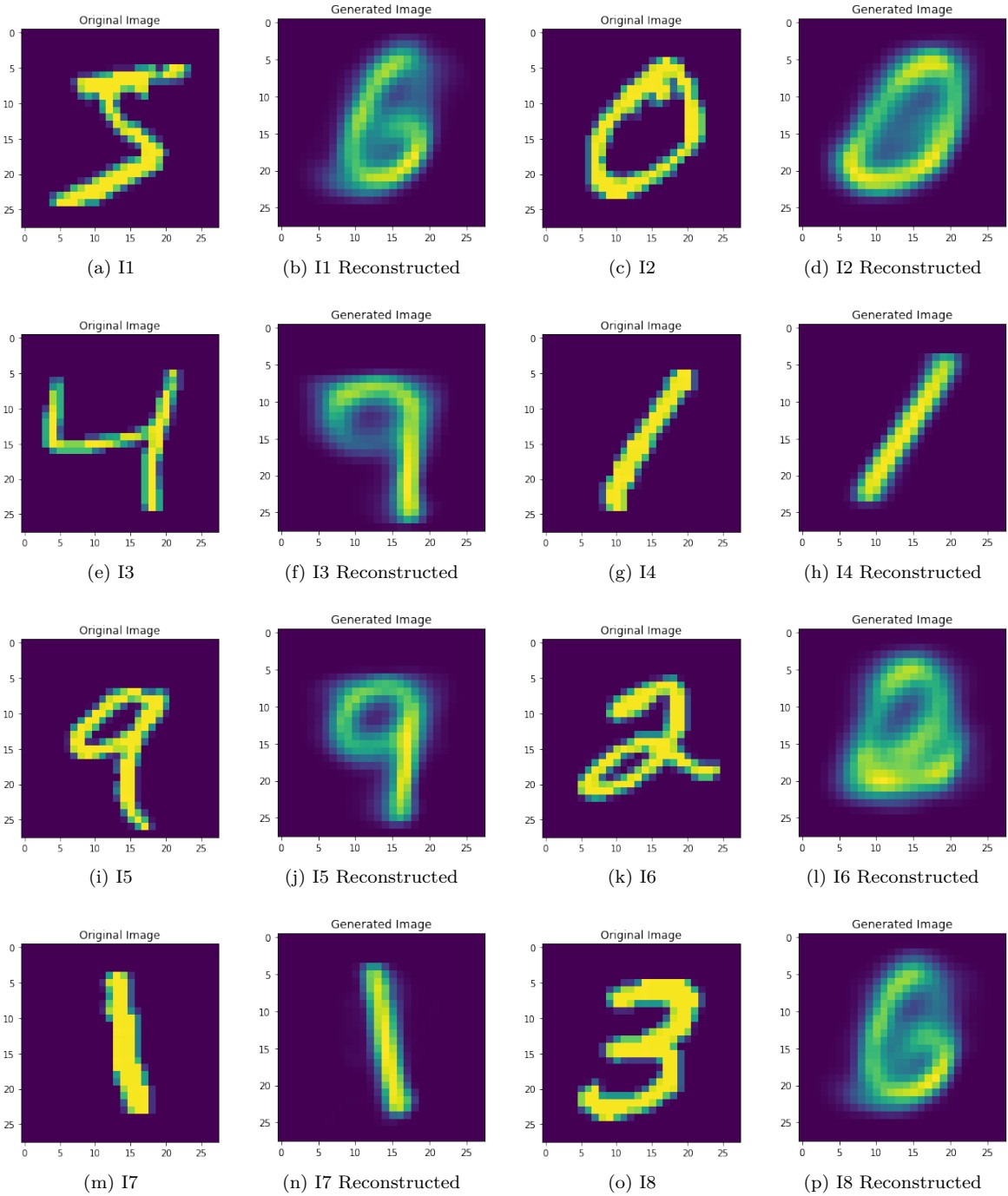

Figure 13: Sampled images and their reconstructed versions using the trained MNIST VAE

numbers did not seem to be very separated in latent space. It was not feasible to look at all the images to qualify results, but the loss function did converge to a much lower value, since the generative loss function is much lower. The KLD loss was higher $\approx 950$ vs. $\approx 660$ for the VAE model. We seemed to get fewer obvious mistakes from the VLT, although there were some issues with confusing numbers as before, albeit less in a random sample. See Figure 18 for the latent space, and Figure 19 for some example images and their reconstructions.

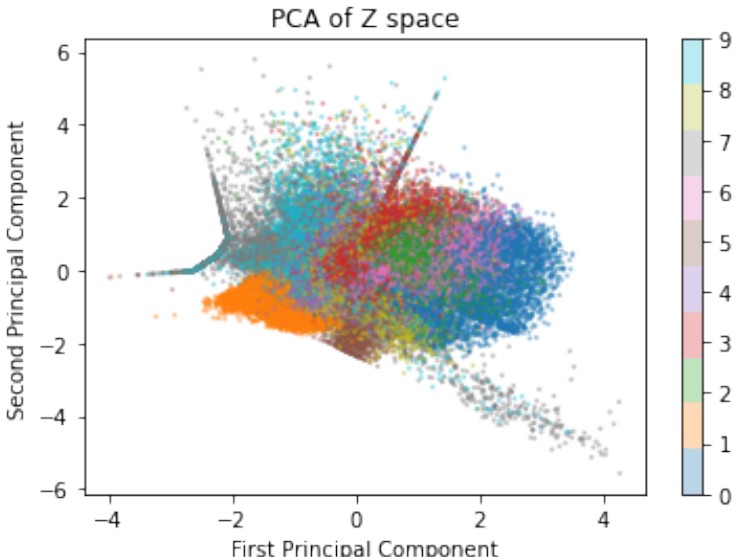

Figure 14: Shift1: The latent space of test images of the MNIST VAE, test images are original images shifted by 1 pixel to the right, with the right most pixel rolled over to the left most column.

## C   Cryo-EM Experiments

All the experiments presented used the EMPIAR-10076: L17-Depleted 50S Ribosomal Intermediates. This dataset was ~130,000 images, downsampled to 128x128 images, with about $\sim 1.3$Å/pixel resolution.

Figure 20 shows two example volumes from the tutorial exercise for CryoDRGN.

### C.1   Variational Lookup Table Experiments

In this section we discuss briefly modifications of the Variational Lookup Table experiment in Section 3.2. These additional experiments showed qualitative similarity to the standard CryoDRGN VAE.

#### C.1.1   Optimizing the Latent Space Embedding

In this experiment we initialized the means $\mu_i$ in the VLT table to zero (rather than a normal distribution), and the log variance $\ln(\sigma_i)$ to $-8$ (standard CryoDRGN runs typically converge to values in the order of $\ln(\sigma_i) \approx -5$ by the end of the run). Other than the initialization, the optimization followed the same procedure at the VLT discussed in the main text. The results of this experiment are presented in Figure 21.

#### C.1.2   No Optimization of the Latent Space

In the three experiments in this section we fixed the means $\mu_i$ and the log variance $\ln(\sigma_i)$ in the VLT table and optimized only the decoder.

First, we set the values of the $\mu_i$'s in VLT table to zero and optimized the decoder alone. As expected, the trained encoder produced a single volume that reflects some form of an averaged consensus, presented in Figure 22. Next, we assigned the means $\mu_i$ random values drawn from a normal distribution. Again, we optimized only the decoder, which produced small variability in volumes; a representative volume is presented in Figure 23a. The values of the latent variables are not updated during the run, so they are of course simply the initial values, for consistency we illustrate these values as in other experiments in Figure 23b.

Finally, we fixed the values in the table to those produced by the CryoDRGN encoder in the baseline experiment in Figure 3. We reset the *decoder* parameters to random values and optimized only the decoder

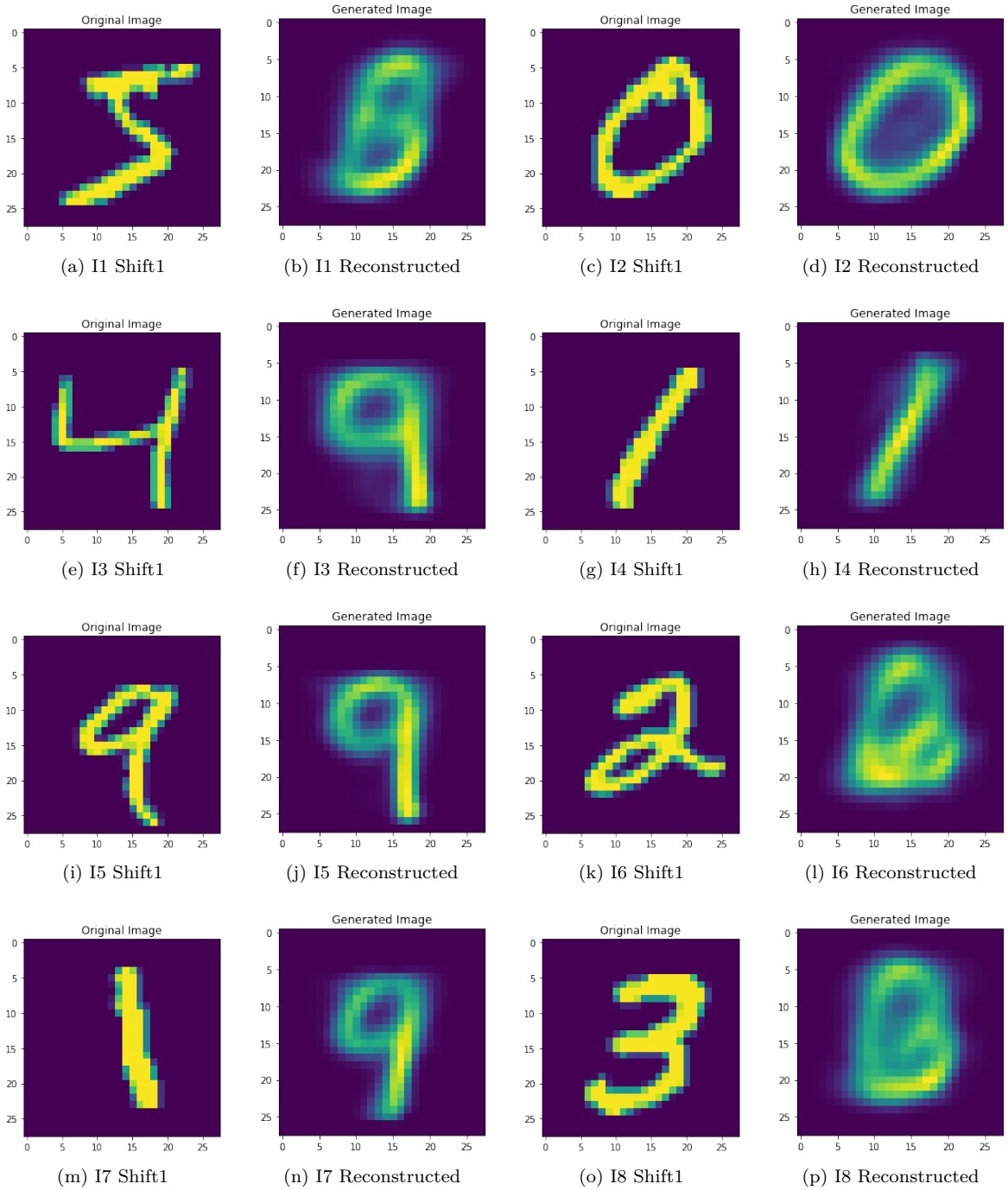

Figure 15: Images shifted by 1 pixel and the reconstructions using the trained MNIST VAE

with the fixed values of latent variables. The results, presented in Figure 24, are rather similar to those produced in the baseline.

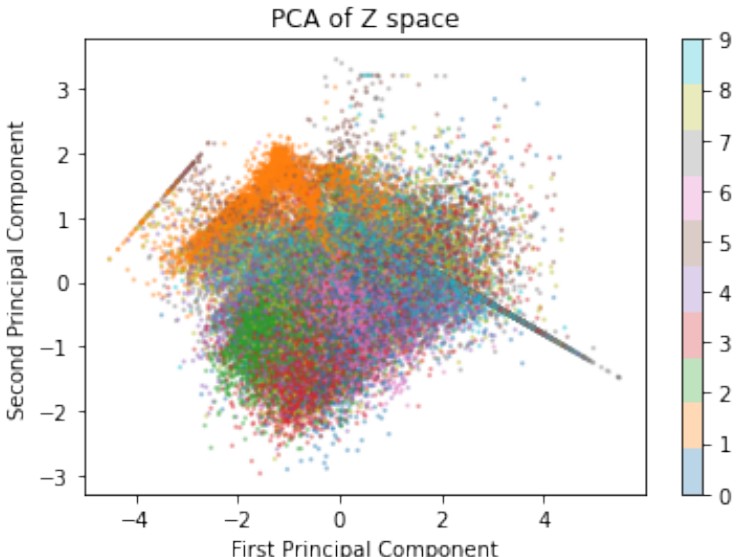

Figure 16: Shift5: The latent space of test images of the MNIST VAE, test images are original images shifted by 5 pixels to the right, with the 5 right-most columns rolled over to the left most columns.

## C.2 CryoDRGN with a Smaller Encoder and Latent Space

We additionally qualitatively tested the effects of the encoder size and the latent space dimensionality on the results of the CryoDRGN model. In the following experiment, the encoder is a 2 layer, 256 wide network and the latent space is 2, down from a 3 layers, 256 wide network with an 8 dimensional latent space in the baseline experiment. We present the results in Figure 25.

The conformational heterogeneity is more difficult to identify in this experiment, and the clustering is less informative. However, significant informative conformational heterogeneity is visible in the figure.

## C.3 Smaller Encoder, Evil Twin

We repeated the experiment with the smaller network and smaller latent space from Section C.2 in the evil twins setup from Section 3.3, using the permuted images as the evil twin pairing. The results are presented in Figure 26.

## D Computation time

All experiments were run on the NVIDIA Quadro RTX 4000 GPU using 4 Intel(R) Xeon(R) Gold 6254 @ 3.10GHz cores with 80GB of memory allocated. All experiments with the cryo-EM datasets were run for 50 epochs. Running longer did not appear to qualitatively improve the results as loss functions appeared to qualitatively converge around this 50 epoch cutoff. This is also the number of epochs recommended for training in the CryoDRGN tutorial. We report the average time per epoch of representative experiments in Table 1. The run times for the different variations of CryoDRGN presented in this paper are very similar.

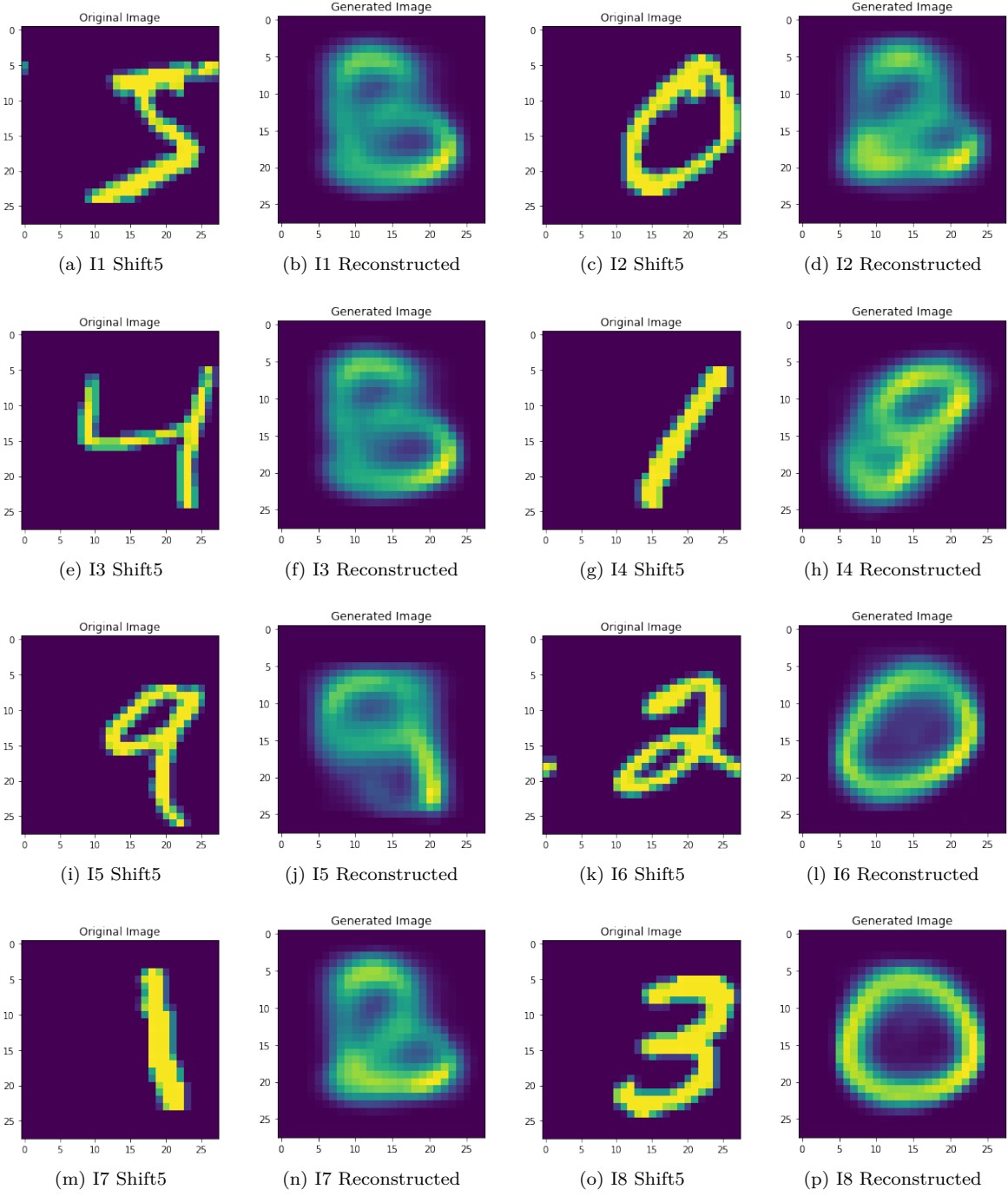

Figure 17: Images shifted by 5 pixels and the reconstructions using the trained MNIST VAE

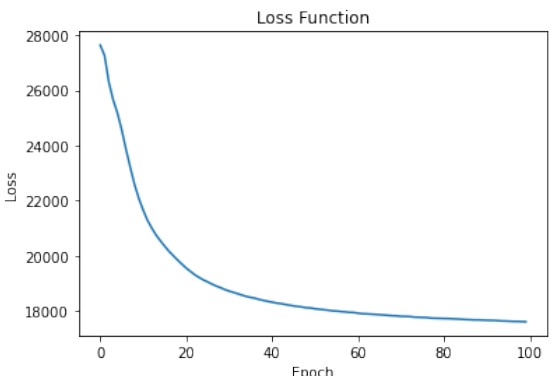

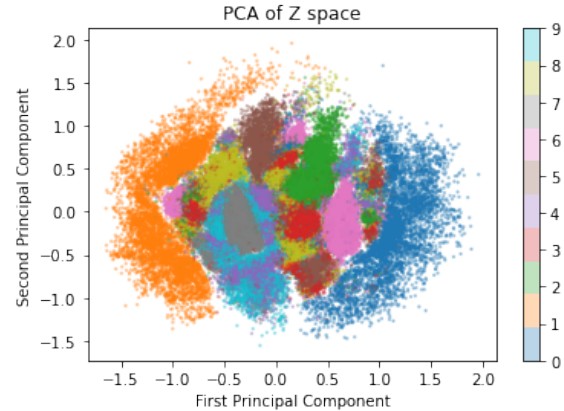

(a) MNIST VLT loss function, run to convergence after 100 epochs

(b) 2D Latent space learned by the MNIST VLT, the latent space is in 2D so the principal components are equivalent to the dimensions. Colored by labeled digits

Figure 18: The MNIST VLT

Table 1: Select experiments and average time to train the associated network per epoch (reported by CryoDRGN).

| Experiment | Time per Epoch (mm:ss) |
|---|---|
| Baseline Tutorial (Figure 3) | 09:57.79 |
| VLT, Random Initialization (Figure 5) | 09:59.79 |
| Evil Twin, Permuted (Figure 7) | 10:02.35 |
| Evil Twin, Permuted, Large (Figure 8) | 09:27.05 |
| Evil Twin, Permuted, Small (Figure 26) | 09:57.79 |

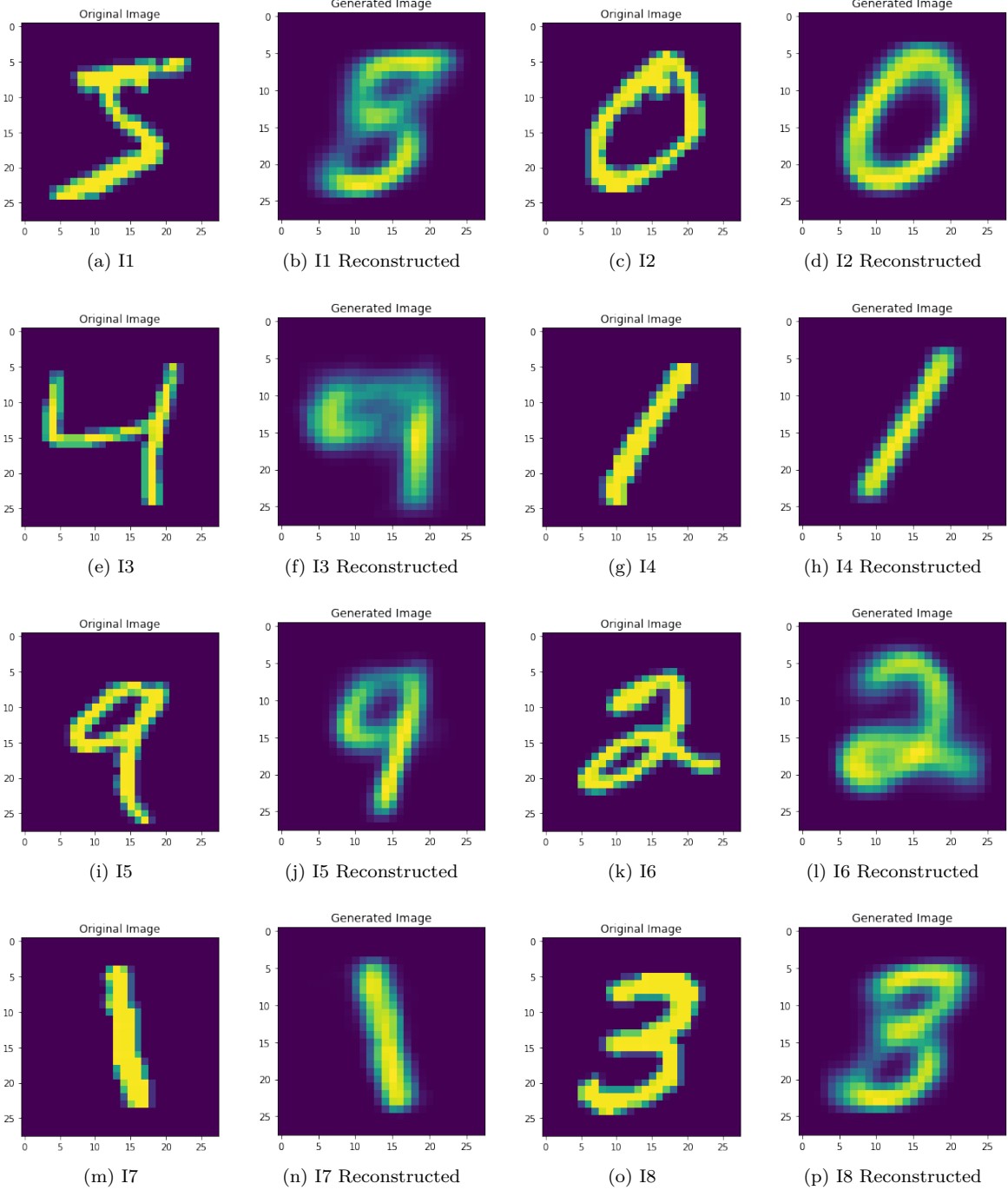

Figure 19: Sampled images and their reconstructed versions using the trained MNIST VLT

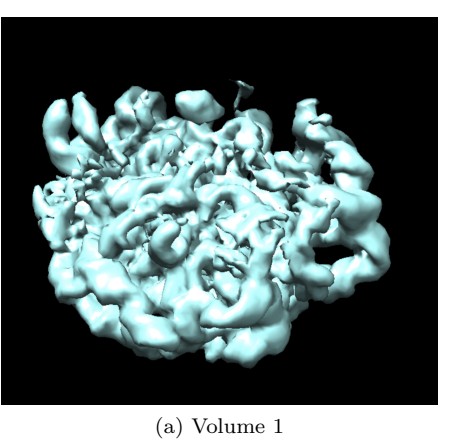
(a) Volume 1

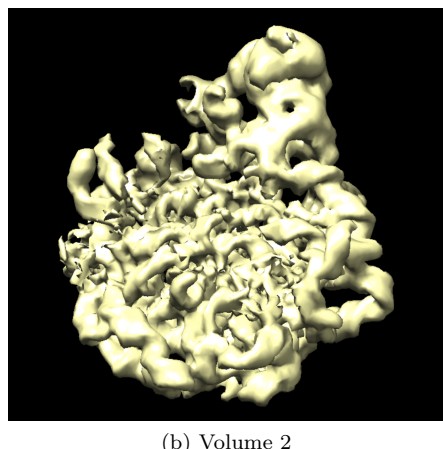
(b) Volume 2

Figure 20: Volumes generated from using images 1 and 2 as generating images for volumes produced from the basic CryoDRGN training procedure.

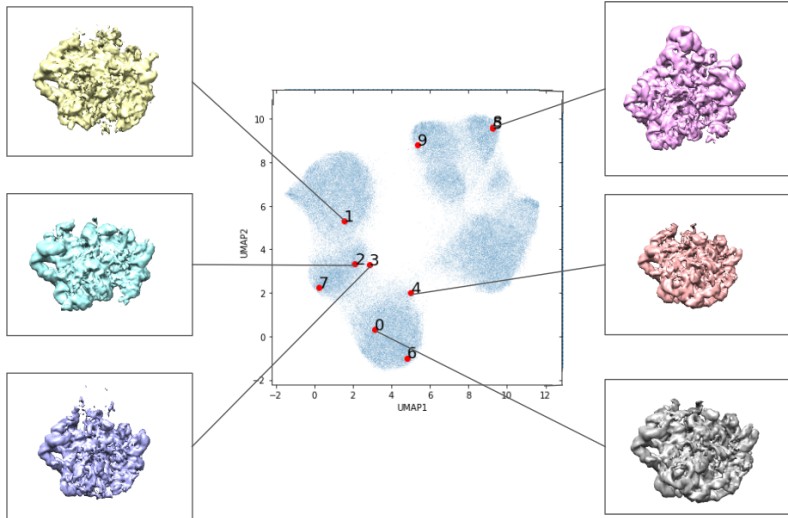

Figure 21: Zero initialization of the latent space using the Variational Lookup Table architecture with latent space optimization. Labeled locations in the learned latent space of the first ten images in the data set visualized with UMAP dimensionality reduction. Volumes associated with certain areas of the latent space are included.

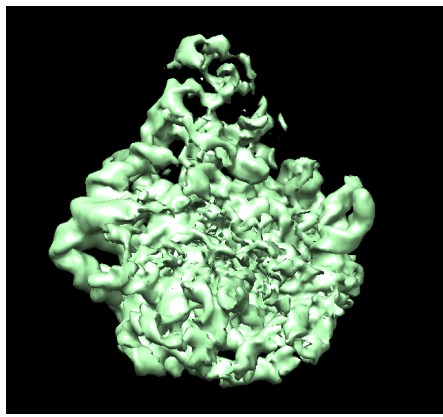

Figure 22: The only volume produced by the VLT with all points initialized to zero, with no optimization on the latent space.

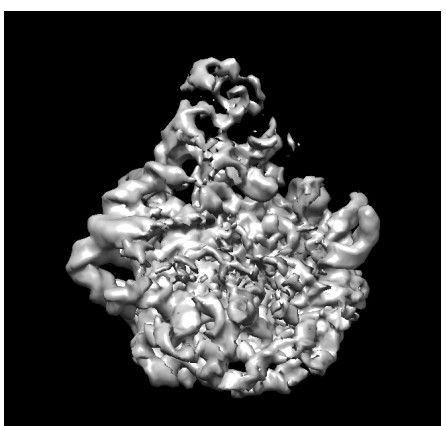

(a) The only unique volume produced by the VLT with random initialization, with no optimization on the embedding layer

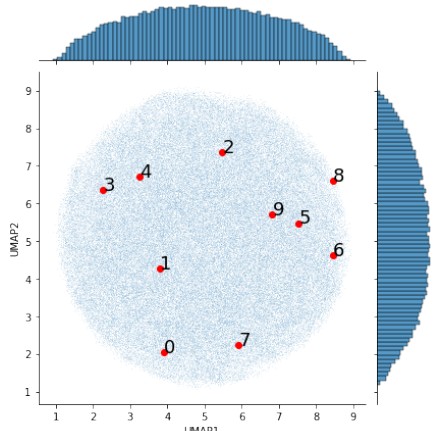

(b) VLT, Random Initialization, No optimization on embedding layer, UMAP.

Figure 23: The VLT with random initialization, no optimization on the latent space.

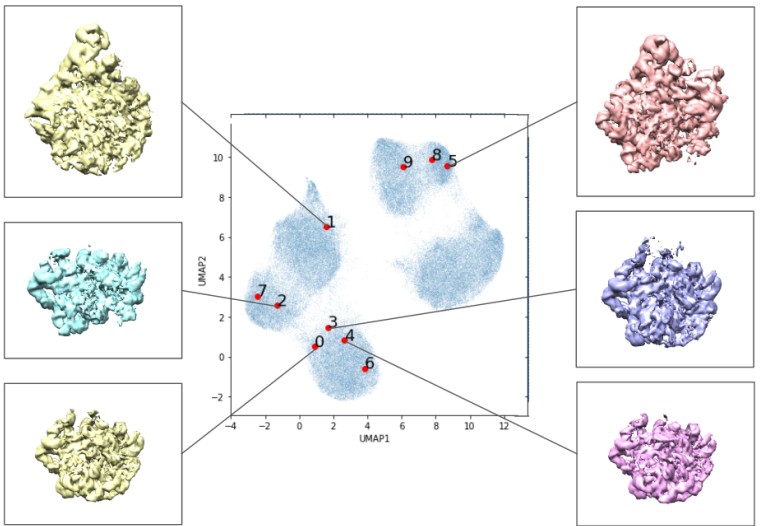

Figure 24: Pre-trained initialization of the latent space using the Variational Lookup Table architecture with no latent space optimization. Labeled locations in the learned latent space of the first ten images in the data set visualized with UMAP dimensionality reduction. Volumes associated with certain areas of the latent space are included.

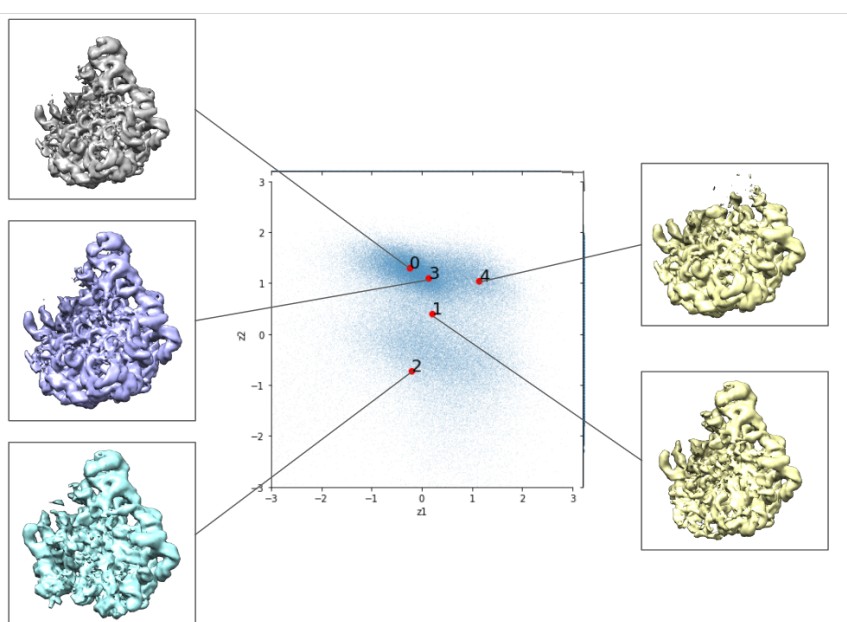

Figure 25: The standard VAE experiment with a small encoder and latent space ($z \in \mathbb{R}^2$). Labeled locations in the learned latent space of the first five images in the data set visualized in the 2-dimensional latent space. Volumes associated with certain areas of the latent space are included.

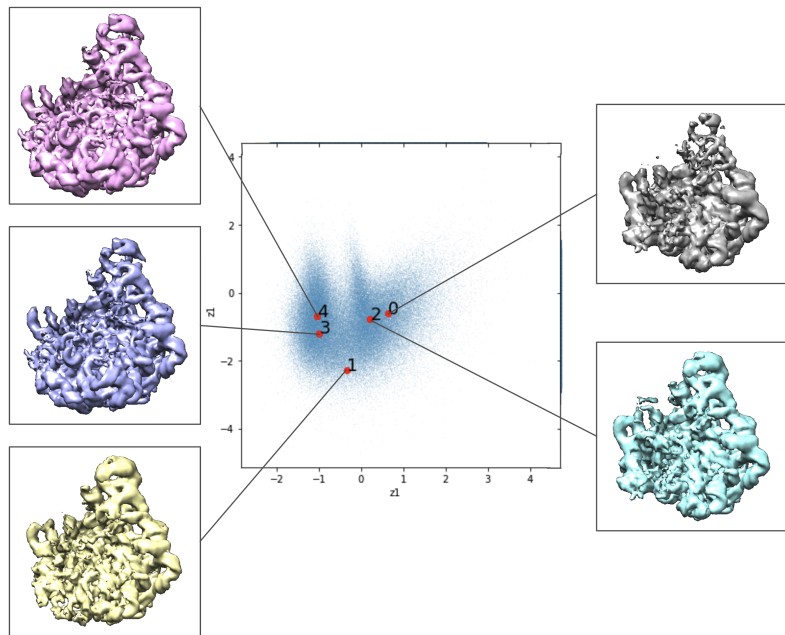

Figure 26: The evil twins experiment, permutation variant, with a smaller encoder and latent space.

