# OpenReview forum: "Using VAEs to Learn Latent Variables: Observations on Applications in cryo-EM"
_TMLR — Rejected by TMLR_

### Review · Reviewer_WY7v · 2023-07-17

**Summary Of Contributions:**

This paper takes a specific example of a VAE trained on cryo-EM data and aims to test the hypothesis that the encoder generalizes well to unseen data. The authors provide several experiments which they claim suggest that the amortization that the encoder uses is unhelpful for performance, that the encoder is able to overfit to randomly labelled data, and that the encoder does not generalize well to transformations of the original dataset.

**Audience:**

Yes

**Claims And Evidence:**

No

**Requested Changes:**

My main requested change would be clarification as to whether the experiments the authors provide are sufficient to establish their claims. If not, then additional experiments would be required in order to substantiate the core claims of the paper. Some comments I have on the current set of experiments are:
- The authors only ever evaluate their results qualitatively, via UMAP visualisations. As far as I can see, these provide a broad indication as to how latent variables are clustered. But, I don’t think that two UMAP visualisations which are perceptually similar provide compelling evidence that the underlying latent variables have the same structure, certainly not to the extent required to make the claim that e.g. the cryoDRGN and VLT models have equivalently performing latent representations.
- In addition, while the authors do acknowledge the lack of quantitative measures, I don’t see how they can provide sufficient evidence for their claims in the absence of such quantitative measures.
- Relatedly, the discussion of the results in Section 3.2 seems lacking; the authors say little beyond asserting that the VLT results are qualitatively similar to the original model. From this alone, it is unclear what we should infer.
- In Section 4, when talking about the experiments of Section 3.2, the authors imply that they have shown a similar performance between the VLT and VAE models. But as far as I can see, they've only provided a visualisation that the latent embeddings are qualitatively similar. This seems very different from the claim that the two models have similar reconstruction loss, which is what I would naturally assume Section 4 was talking about.
- In Section 3.3, it is not clear what the authors mean by the "conformational variability" or how they judge it to be similar to the baseline. Further, there is little discussion of the interpretation of the results here or in Section 4. The only claim I see made is "encoder can effectively overfit the values of the latent variable", but this seems to be relatively trivial, and distinct from the claim the authors would like to make, which I understand as being something like "the model trained on the true labels only succeed because it is memorizing the training data".
- In Section 3.4, you claim that the model doesn’t work on translations or rotations of the dataset images. However, the evidence you provide for this claim is that the latent representations of the transformed inputs are different in the UMAP projections. This doesn't seem to be significant evidence for the claim you are making; unless I'm missing something, the model could still achieve a very good reconstruction loss even if there were several points in the latent space all corresponding to the same biological structure, so long as the decoder mapping was many-to-one. Further, there seems to be no comment on the reconstruction loss of the transformed inputs, and I would assume this is the main way to measure the success of the model on the transformed data. Am I misunderstanding? From the presented evidence, it seems to me entirely plausible from the evidence shown that the model does in fact work for the translations.

Beyond the clarifications above, I have some further minor comments:
- The terminology “explicit representations” seems a bit misleading. As I understand it, by explicit latent representations you mean the latent representations that are learned via non-amortized VI. The use of "explicit" here threw me and might be worth changing.
- I found the sentence “The amortization gap (Cremer et al., 2018) … with respect to the ELBO” a bit unclear and had to go back to the original paper to work out what you meant. This might be worth rephrasing.
- I had to look up the terms “tomographic projection” and “contrast transfer function” as I was unfamiliar with them, and I expect other readers may be too. They might be worth defining.
- With regards to the experiments in Section 3.3, there has been a decent amount of prior work regarding the ability of neural networks to fit random labels (see e.g. https://arxiv.org/pdf/2006.10455.pdf and some of the related work). It would be nice to reference some more of this work and discuss your results in the context of previous findings.

**Strengths And Weaknesses:**

### Strengths

- The background provided on VAEs is clear and easy to understand for the most part, and the background on cryo-EM is useful for a reader without familiarity with biological applications of machine learning methods.
- The set-up for the experiments is clearly described, and the results that the authors do have are clearly presented.

### Weaknesses

- I am unconvinced that the results of the experiments in the paper are sufficient to establish the claims that the authors make. I find the experiments in Sections 3.2, 3.3 and 3.4 unconvincing for reasons described in more detail below.
- In particular, the lack of any quantitative evaluation really hampers the ability to reliably assess the authors' core claims.
- In addition, I think the authors over-interpret their results in several places, or equivocate between what they hope to prove through their experiments and what the experiments actually show.

---

### Review · Reviewer_YDDg · 2023-09-04

**Summary Of Contributions:**

The authors present a study of the variational posterior in VAEs trained on a Cryo-EM dataset, including qualitative results on the amortization gap as well as overfitting.

**Audience:**

No

**Broader Impact Concerns:**

No concerns

**Claims And Evidence:**

No

**Requested Changes:**

Here I provide more detailed feedback on the paper. I gave most of my feedback on the Methods, Results, and Discussion sections above, so here I include only feedback on the Intro and Preliminaries sections.

### Introduction
* The notation x = {x_i} is needlessly vague. Is i indexing the data points? How many data points are there? I would prefer notation like x = x_1, \ldots, x_N or x = {x_i}_{i=1}^N. Also it seems like x is being used both as “all data points” and “a single arbitrary data point”, i.e. you define x as {x_i}, but then later write q(z \mid x) and p(x, z), which seem to be referring to single data points. Please clarify the notation.
* “They are trained… to approximate the distribution p(x, z)” is too vague, possibly incorrect.
* “Indeed, this generalization is observed in many applications, and the ability of the encoder to compute the latent variables for new unseen data points is used in some applications.” can you provide sources for this?
* Having multiple names for the encoder and decoder is confusing, i.e. the encoder is both Enc_\xi and q_\xi(z \mid x) and the decoder is both Dec_\theta and p_\theta (x \mid z). It would be helpful to define these terms’ relations to each other when they are first used.
* The sentence “In the latter case, the number of variables grows” is confusing. Non-amortized variational inference has most of the same theoretical guarantees that amortized VI does. Please be more specific about your claims on convergence and provide sources.
* It’s not clear what you mean by an ‘explicit variational representation’. Amortized approaches could be thought of as 'explicit' in many ways. Please be more specific.
* You revealed your identity by including the github link. I did not look into your identity further to try and minimize the effect on my review, but please do not do this again. You can include placeholder links during the review phase and replace them with the true link in the camera ready.
* “This work is loosely inspired by…” This sentence belongs in a related works section (which is missing and should be added).

### Preliminaries

* “Maximum-Likelihood Estimation” should not be capitalized
* “This integration in Equation 3 is intractable, and thus it is desirable to have an approximation to the distribution p_\theta(z \mid x)” Why is the integration intractable? Why does having an approximation to the posterior solve that problem?
* “We intend to optimize these parameters to best approximate q(z \mid x) \approx p_\theta(z \mid x)” is grammatically incorrect. Additionally, please be more specific and describe in what sense q is approximating p.
* The ELBO is just the “evidence lower bound”, not the “evidence-based lower bound”.
* “In many applications the amortized learning of shared encoder variables reduces the computational cost” Please be more specific, this is only true of memory. Total flops are generally reduced by using a non-amortized posterior as it doesn’t need to run a neural network forward or compute its gradients at each step.
* What is the “true distribution” p^*(z \mid x) and how is it different than the posterior p_\theta(z \ mid x)
* You state that “V: R^3 -> R” is the “3D structure we want to estimate”. This is confusing because V is the electron density and is a map from all of R^3 to the real line. Please clarify how this is discretized or represented in the network. More generally, your explanation of the underlying model is vague. Expanding it would improve the work.

**Strengths And Weaknesses:**

## Strengths

The problem of inferring 3D structures of molecules is important, and the qualitative results on generalization of the encoder directly highlight a real issue.

## Weaknesses

The first criteria for acceptance to TMLR is “Are the claims made in the submission supported by accurate, convincing and clear evidence?”. Unfortunately, this paper does not support its claims with convincing or clear evidence.

### Amortization

The first claim the authors make is to investigate the “amortization gap” of a VAE applied to cryo-EM data. To investigate this, they fit both amortized and non-amortized VAEs to the data, and compare the resulting latent spaces qualitatively by examining UMAP embeddings of the latents.

It is not possible to draw any firm conclusions about the amortization gap from this experiment. You claim there is an “absence of established quantitative methods for comparison of heterogeneous structure analysis”, but it should be possible to directly compute the amortization gap: compute lower bounds on the marginal likelihood assigned to the training set under amortized and non-amortized VAEs. Why was this not done?

I don’t see how it is possible to even draw broad qualitative conclusions about the latent space from these experiments. Some issues I have are:

* Different settings of the UMAP algorithm hyperparameters can give vastly different results, and the hyperparameter settings were not given in the paper.
* The algorithms under study are random, and the authors do not present results from different seeds so we do not know how much the results are due to chance.
* The authors only present the locations of 10 data points, limiting my confidence about the generalizability of the results.
* The authors seem to be operating under the assumption that the latent space is smooth, or that similar molecules should be embedded near each other. That is not necessarily the case – the complexity of the neural network decoder means that points close to each other in latent space could represent molecules humans would consider highly dissimilar.
* The authors do not at all consider the variance of the Gaussian produced by the variational posterior. It is possible that the variances are broad and thus the means of the variational posteriors are less meaningful.

Overall, the authors provide evidence that 10 data points are embedded near each other by both amortized and non-amortized VAEs when the latent space is transformed under UMAP. In my opinion this is not enough evidence to draw conclusions about the amortization gap of these VAEs. I also do not understand why characterizing the amortization gap for these VAEs is important in the absence of quantitative or qualitative results showing lacking performance.

### Generalization

The authors also claim to investigate the generalization of the approximate posterior, but confusingly these results are under a section labeled “3.4 Amortization Experiments”.

The results presented in this section are more convincing than the previous section, but they suffer many of the same issues — UMAP is used which needlessly obscures the results, only 10 points are shown, no quantitative metrics are given, no uncertainty is quantified. It is interesting that the corrupted images tend to be encoded to different locations than the uncorrupted images, but there is nothing in the architecture of the encoder network or its training scheme that would prevent that from happening. For example, in image processing where invariance to translation or scaling is desired, it is common to use convolutional architectures and augment the data with random crops or zooms. In summary, this section verifies issues with the encoder which are not surprising or novel.

Even if the authors had provided clear claims and evidence to support them, I believe that this paper would not rise to the standard of acceptance. The results presented have been seen previously many times in many different contexts (including in several of the papers cited by the authors), and restating them for Cryo-EM VAEs does not, in my opinion, meet the standard of ‘interesting for the TLMR audience’.

To be an interesting paper, I would have expected the authors to propose and test fixes to the issues with CryoDRGN.

---

### Review · Reviewer_GxUB · 2023-09-11

**Summary Of Contributions:**

This work studies the amortization properties of VAEs in a case of a biological application cryo-EM. The authors work on an existing VAE framework CryoDRGN, and perform detailed case study on various directions of it. The authors first test the performance of the un-amortized VAE (i.e. VLT in the paper) and found this version perform (qualitatively) close to the amortized version. Then, an experiment on randomized testing data show that VAE will overfit on the data and do not generalize well. In addition, the authors perform several small variations on the image (e.g. rotation, translation) when testing the VAE performances and argue the studied VAE does not generalize well even on these augmented data hence VAE can not generalize well on general unseen data for this application.

The authors also test their ideas on public datasets like MNIST, but get different conclusions and they present the reasons for it. The authors also clearly discuss about the limitations (mostly related to data) of the proposed idea.

**Audience:**

Yes

**Broader Impact Concerns:**

No concerns for this work.

**Claims And Evidence:**

No

**Requested Changes:**

1. Perform studies on more cases on cryo-EM on more datasets. Show quantitative results and theoretically analyze the propositions.

2. If possible, try to find out a solution for the generalization issue. This work will make more contributions if this issue can be fixed (or at least partially fixed).

**Strengths And Weaknesses:**

Strengths:

1. The experiments are well designed. We can clearly understand the authors' purpose and it gives us some reasonable conclusions on the generalization properties of VAE.

2. This paper is well-written and easy to follow. The visualizations are great for people to understand the results, even if they are not familiar with the background of the application.

Weaknesses:

1. Perhaps I am not familiar with cryo-EM and not familiar with its current state-of-the-art modeling ideas. But I am concerned about the innovation and contribution of this work.

From a modeling perspective, VAE is a well-known model and the conclusions that the authors got are not innovative. The authors claim that VAE may overfit when the network is big and it might not generalize well on some unseen data. These are correct but they are not some new findings. VAE is a supervised learning idea and it is totally possible to overfit, and perform bad on unseen data with a distribution far from that of the training data. The authors also mention that VAE cannot generalize well when they use rotated/translated images in testing. But that is also not hard to understand without an experiment and we cannot say that they will generalize well to such examples unless we put specific efforts on the loss function or training data (e.g. adding some randomly transformed images to training as well).

From an application perspective, again I am not familiar with cryo-EM but definitely there have been many algorithms on it. This paper is not the first paper to apply a new idea to this domain. It is pointing out some issue of an existing idea (CryoDRGN) but this issue is not a general issue for most common ideas in VAE for cryo-EM. I am concerned if the contribution of this work is sufficient for a paper appearing at TMLR.

2. The major design for the experiments are good. But some of the details might be doubtful. For example, the authors only study one VAE setting (CryoDRGN) for cryo-EM, which might not be sufficient to propose a general conclusion. In addition, the major part of this paper is based on one single real dataset, which is not sufficient. Another one is that the authors mention that the reason why they do not have quantitative results is because such metrics are hard to get/define. But definitely there are some basic metrics (e.g. reconstruction error) that we can look into. Purely seeing qualitative results would not be convincing. Moreover, the authors always use the first 10 examples in the dataset to show the results. Though I have not looked at the data, I highly doubt using the first 10 data points would lead to a biased conclusion. Perhaps a fixed set of 10 randomly chosen data points would be a better choice.

3. This work is lack of quantitative and theoretical analysis on the conclusions that are proposed. It is better if the authors can analyze the conclusions quantitatively instead of only performing some experiments.

---

### Decision · Action_Editor_hLbL · 2023-12-10

**Recommendation:** Reject

**Comment:**

The paper has been reviewed by 3 reviewers. Two reviewers (YDDg & WY7v) recommended rejection while the other one (GxUB) never submitted their final recommendation.

The paper addresses an interesting problem: consider a VAE trained on cryo-EM data. It aims to test the hypothesis that the encoder generalizes well to unseen data. The background on VAEs and cryo-EM is nicely written.

However, as mentioned earlier, many criticisms were raised by reviewers YDDg and WY7v. The authors make many claims which do not appear quite supported by their results. The rebuttal didn't change their opinion with Reviewer WY7v stating "The authors have replied to the comments of the three reviewers, but I feel that their replies have either missed or deliberately avoided the point of the criticisms in a lot of places. Most of the original criticisms, in particular the concerns about novelty and the validity of experiments, therefore still stand.' and Reviewer YDDg  stating "I thank the authors for their response, but my opinions of the paper remain mostly unchanged. The paper would need to be substantially reworked to support its claims with compelling evidence. I also think interest to the TMLR audience is still marginal as the authors did not propose any solutions to the problems they highlight."

I therefore recommend rejection of the manuscript.

**Audience:**

The paper investigates the performance of VAEs for Cryo-Electron Microscopy. Some individuals in TMLR's audience would be interested in the topic. Unfortunately the paper would need to be bringing new compelling evidence to better support its claims.

**Claims And Evidence:**

There are a number of claims in the paper that do not appear to be substantiated by the results. This point was raised very explicitly by two reviewers   (YDDg and WY7v):

"The first criteria for acceptance to TMLR is “Are the claims made in the submission supported by accurate, convincing and clear evidence?”. Unfortunately, this paper does not support its claims with convincing or clear evidence." Reviewer YYDg

"I am unconvinced that the results of the experiments in the paper are sufficient to establish the claims that the authors make.", "In addition, I think the authors over-interpret their results in several places, or equivocate between what they hope to prove through their experiments and what the experiments actually show."  Reviewer WY7v

After a careful reading of the manuscript, I have to concur with the reviewers, the paper does not provide convincing evidence for many of the claims made.  The reviewers were not swayed either by the rebuttal which does not address convincingly many of the issues raised.